# Nanoencapsulation of *Vaccinium ashei* Leaf Extract in Eudragit^®^ RS100-Based Nanoparticles Increases Its In Vitro Antioxidant and In Vivo Antidepressant-like Actions

**DOI:** 10.3390/ph16010084

**Published:** 2023-01-07

**Authors:** Verciane Schneider Cezarotto, Eduarda Piovesan Franceschi, Ana Cristina Stein, Tatiana Emanuelli, Luana Haselein Maurer, Marcel Henrique Marcondes Sari, Luana Mota Ferreira, Letícia Cruz

**Affiliations:** 1Programa de Pós-Graduação em Ciências Farmacêuticas, Departamento de Farmácia Industrial, Universidade Federal de Santa Maria, Santa Maria 97105-900, RS, Brazil; 2Departamento de Ciências da Saúde, Curso de Farmácia, Universidade Regional Integrada do Alto Uruguai e das Missões, Frederico Westphalen 98400-000, RS, Brazil; 3Programa de Pós-Graduação em Ciência e Tecnologia dos Alimentos, Departamento de Tecnologia e Ciência dos Alimentos, Centro de Ciências Rurais, Universidade Federal de Santa Maria, Santa Maria 97105-900, RS, Brazil; 4Departamento de Farmácia, Universidade Federal do Paraná, Curitiba 80210-170, PR, Brazil

**Keywords:** blueberry, leaves hydroalcoholic extract, depression, polymeric nanoparticles

## Abstract

Depression is a major psychiatric disorder in Brazil and worldwide. *Vaccinium ashei (V. ashei*) leaves are cultivation by-products with high bioactive compound levels. Here, a hydroalcoholic extract of *V. ashei* leaves (HEV) was associated with Eudragit^®^ RS100-based nanoparticles (NPHEV) to evaluate the in vitro antioxidant and in vivo antidepressant-like effects. Interfacial deposition of the preformed polymer method was used for NPHEV production. The formulations were evaluated regarding physicochemical characteristics, antioxidant activity (DPPH radical scavenging and oxygen radical absorbance capacity), and antidepressant-like action (1–25 mg/kg, single intragastric administration) assessed in forced swimming and tail suspension tests in male Balb-C mice. The NPHEV presented sizes in the nanometric range (144–206 nm), positive zeta potential values (8–15 mV), polydispersity index below 0.2, and pH in the acid range. The phenolic compound content was near the theoretical values, although the rutin presented higher encapsulation efficiency (~95%) than the chlorogenic acid (~60%). The nanoencapsulation improved the HEV antioxidant effect and antidepressant-like action by reducing the immobility time in both behavioral tests. Hence, Eudragit^®^ RS100 nanoparticles containing HEV were successfully obtained and are a promising alternative to manage depression.

## 1. Introduction

Depression is a common psychiatric disorder that is often neglected, especially among individuals living in socioeconomic vulnerability situations [1]. According to the World Health Organization (WHO), 3.8% of the population is affected by depression, including 5.0% among adults and 5.7% among adults older than 60 years, which represents approximately 280 million people in the world [2]. The global burden of depression cases is in low- and middle-income countries, where health systems are not able to meet the high demand and there are scare economic resources [1,3,4]. The prevalence of mental diseases is elevated in Brazil due to the emerging economy with high social disparities. Evidence demonstrated that depression occurrence is mainly associated with demographic, economic, social factors, and neighborhood characteristics. In 2015, 5.8% of the Brazilian population suffers from depression [4]. Remarkably, 10.8% of adults were depressed in 2019, but over 70% of them did not receive care [3]. Additionally, in March 2020 the WHO declared the COVID-19 pandemic, leading social isolation and distancing, online learning, loss of jobs, and difficult access to health services, which increased depression incidence [5]. 

In view of this scenario, there is a concern about the increasing antidepressant uses due to unanswered questions regarding their effectiveness and safety [6]. Currently, there are more than 20 antidepressant drugs in 780 medicines available in Brazil [6]. Although, limited efficacy, pronounced delay to action onset, and undesirable effects (hypertensive crisis, sexual dysfunction, body weight change, and sleep disturbances) are observed in patients [6,7]. In this context, plants are natural sources that could provide a safer and more effective therapy than conventional pharmacological treatments. Notably, the scientific literature has reported that natural products present psychotherapeutic potential in a variety of preclinical models [8,9,10,11,12]. 

Blueberries (*Vaccinium* spp.) are small fruits that have been receiving special attention among the ones cultivated in Brazil because their economic value and nutraceutical properties [13]. Blueberry cultivation was recorded in Rio Grande do Sul in the decade of the 1980s and the current area of blueberry cultivation in Brazil is estimated at approximately 400 hectares [14]. *Vaccinium ashei* (*V. ashei*) is a fruit tree and is recognized for its high bioactive compounds and health benefits obtained by the consumption [15,16]. While the main commercial interests are focused on the fruits, blueberry leaves are by-products of berry cultivation that have been shown high phenolic compound levels (e.g., chlorogenic acid [CGA] and rutin [RU]—Figure 1), which provide to the extract antileukemic, antihypertensive, hypolipidemic, and neuroprotective actions as well as atherosclerosis and cancer prevention [17,18,19,20,21,22,23]. Previously, the effect of harvest season variation and cultivar on the phenolic and flavonoid total content, phytochemical profile, and antioxidant action of *V. ashei* (Rabbiteye blueberry) hydroethanolic leaf extracts (HEV) grown in Brazil was demonstrated [17]. 

However, despite the biological potential of the substances in vegetal matrixes, variable physicochemical characteristics can be observed, which affect the bioavailability and limit the biological action [24,25]. Natural extracts are still poorly clinically used because their highly water solubility hinders phytochemicals absorption, as they are unable to permeate the lipid membranes of cells [26]. Furthermore, phytochemicals may have an inadequate molecular size and physicochemical instability, requiring high concentrations in biological systems for an effective pharmacological action [12,26,27,28]. Thus, to plan adequate alternatives to overcome these restrictions and enable to applicate plant extracts for different purposes, including the mood disorders treatment, is imperative. Contemplating a strategic tool for obtain platforms for the bioactive compounds incorporation, the development of nanotechnology-based formulations stands out [26].

Nanoscience is an area with great potential for technological applications, where nanotechnology is considered a new complementary tool for the improvement and development of products [29]. Nano-based formulations present some advantages given the nanometric size and physicochemical characteristics of carriers, such as improved stability, solubility and bioavailability of phytochemicals, controlled release of plant active constituents, decreased side effects, reduced required dose for promoting biological effects, and enhanced pharmacological action [26,30,31]. Given the above, the potentialities of *V. ashei* leaves and the promising association with nanocarriers compelled us to develop polymeric nanostructures containing *V. ashei* leaves hydroalcoholic extract (NPHEV) and to evaluate the in vitro antioxidant effects and in vivo antidepressant-like potential. 

## 2. Results and Discussion

### 2.1. Nanoparticle Preparation and Characterization

The feasibility of encapsulating HEV in Eudragit^®^ RS 100-based nanocarriers was presented here. The formulations were physiochemically adequate and increased antioxidant action. In addition, behavioral findings for the antidepressant-like effect elicited by a single NPHEV administration to animals were provided.

Nanostructures have attracted attention given their efficient active ingredients delivery and fractions extracted from plants [26,28,29]. In recent decades, numerous nanotechnology-based systems have been used for herbal drug nanoencapsulation; for instance polymeric nanoparticles, metal and inorganic nanoparticles, solid lipid nanoparticles, polymeric and phospholipid micelles, and liposomes [29,32]. The polarity, solubility, and volatility of active compounds, and organic solvent presence in the extract are factors that must be considered to select the nanosystem and its preparation technique [27,29,32]. Thus, due to the HEV compounds’ solubility (i.e., poorly soluble in water, soluble in ethanol, and insoluble in ethyl acetate), the polymeric nanoparticle was based on Eudragit^®^ RS100 and prepared by the nanoprecipitation method. In this sense, cationic nanoparticles based on Eudragit^®^ RS100 polymer present advantages regarding mucoadhesion and sustained drug release, which can also improve the pharmacological actions of active substances [33].

After preparation, nanoparticles at 5 and 10 mg/mL of HEV concentration (NPHEV-5 and NPHEV-10, respectively) had a homogeneous and milky appearance without any macroscopic precipitation, as expected for colloidal systems. Meanwhile, the nanocarriers with the highest extract concentration (25 mg/mL; NPHEV-25) showed precipitation immediately after preparation, hindering further investigations. The nanoparticle physicochemical characterization is listed in Table 1.

The nanocarriers had average diameters smaller than 200 nm and PDI below 0.2, indicating a narrow size distribution. The extract concentration did not alter these parameters. Particle size and size distribution are the most relevant nanosystem characteristics because such properties influence the in vivo biodistribution, pharmacological and toxicological effects, and particle targeting. Additionally, the formulation physicochemical stability, drug loading, and its release from the particle can also be impacted by particle size [31,34].

The zeta potential values were positive (from +8.8 ± 0.12 to +15.5 ± 8.24 mV), which is expected due to the cationic nature of Eudragit^®^ RS100 [35,36]. This polymer has a positively-charged quaternary ammonium group, promoting an effective nanoparticle adhesion in negatively-charged mucus of the gastro-intestinal tract, extending formulations’ effective residence time in tissues [35]. Furthermore, similarly to other nano-based formulations composed of Eudragit^®^ RS100, the pH values of NPHEV-5, and NPHEV-10 were within the acid range (3.9 ± 0.12 and 3.7 ± 0.10, respectively) [36]. In addition, the HEV phytochemical composition can influence this parameter, whose compounds may contribute to the final formulation pH.

The encapsulation efficiency (EE) demonstrated that almost 100% of RU was entrapped within the particle polymeric matrix (92 and 94% for NPHEV-5 and NPHEV-10, respectively). However, CGA EE was near 50% (54 and 59% for NPHEV-5 and NPHEV-10, respectively) as shown in Figure 2. Beyond that, a similar profile was obtained for the total compound content in which RU had higher levels than CGA in both formulations (Table 1). In polymeric particles, drugs can be associated in different forms, where they may be adsorbed, molecularly dispersed, or retained in the polymeric matrix [34]. In view of this, the difference in EE and total content might be attributed to the CGA lower lipophilicity compared to the RU; the results suggest that the CGA is more externally associated in the polymeric matrix than RU, becoming susceptible to undesirable reactions such as hydrolysis, esterification, and isomerization [37]. Lastly, considering the low water solubility (0.125 g/L) [38] and Log *p* value (0.61) [39] of RU, it is expected that the drug is dispersed in the polymer matrix, which justifies the high encapsulation efficacy observed.

### 2.2. Antioxidant Activities

The antioxidant effect of HEV and both NPHEV formulations (NPHEV-5 and NPHEV-10) was evaluated using DPPH (1, 1-diphenyl-2-picrylhydrazyl) and oxygen radical absorption capacity (ORAC) assays. Overall, HEV encapsulation increase antioxidant activity in comparison to non-encapsulated extract (*p* < 0.05), as depicted in Table 2. Additionally, these data are in accordance with previous studies that demonstrated the *V. ashei* leaf antioxidant effect in similar assays [40,41]. Remarkably, the antioxidant performance is closely associated with CGA and RU chemical structures. Scientific literature demonstrated that the flavonoid antioxidant properties depend on some structural features, such as the hydroxyl groups quantity and location in the molecule, the 2,3-double bond presence in ring C, 3- and 5-hydroxy groups, and the glycosylation model (C-glycosides or O-glycosides) and position [39]. The CGA presents five active hydroxyl groups in its molecule, which readily neutralize free radicals [42], while the 3’ OH group present in RU structure is essential for the antioxidant potential [39]. 

The DPPH radical scavenging and ORAC assays are common tests applied to assess the antioxidant action of nano-based formulations [43]. The radical neutralizing potential is based on the electron-donating property of radical DDPH coming from substances that present several hydroxyl groups in aromatic rings, such as flavonoids. The active substances that are able to donate hydrogen are considered free radical scavengers [43,44]. Meanwhile, ORAC method estimates the property of antioxidants to protect molecule susceptible to a free radical attack [43]. Spectrophotometric tests can hinder results interpretation because of nanoparticles spectral interference. Then, using tests with different mechanisms such as DPPH and ORAC assays would provide greater reliability to the results [45]. 

The higher antioxidant action observed for nanoparticles has already been reported in the scientific literature [45]. Previous data indicated that the nanometric size of the particles could provide superior contact due to the higher surface area between the hydrogen donator and radicals, facilitating the interaction between the hydrogen atom of the radical site and the encapsulated molecules [45]. 

### 2.3. Acute Antidepressant-like Activity

Depression is a psychiatric disorder that causes a huge impairment in the life quality of the affected individuals [1]. The current pharmacological management of this disease presents many limitations [6], which reinforce the urgent need for improved therapeutic options and the rationale of this study. In this sense, the findings demonstrated that the hydroalcoholic extract of a vegetal by-product nanoencapsulated elicited antidepressant-link action in preclinical screening behavioral tests.

In our previous study, the CGA and RU presence in HEV different cultivars was demonstrated [17]. Remarkably, the antidepressant-like effect of these compounds in animal models was already reported [46], leading to the hypothesis that the HEV associated with nanoparticles could enhance its antidepressant-like action. To investigate this hypothesis, acute treatments with HEV or NPHEV-10 were performed in distinct animal groups for investigating the potential antidepressant-like action using behavioral tests. Considering that NPHEV-10 presented the best physicochemical characteristics among the formulations (Table 1), this formulation was used in animal tests. Different groups of animals were used for assessing dose–response curves for HEV (10, 25, and 50 mg/kg, intragastric administration) or NPHEV-10 (1, 2.5, 5, 10, 25, and 50 mg/kg, intragastric administration). HEV and NPHEV-10 were diluted to achieve the desired concentrations prior to injection in the animals. 

None of the HEV tested doses significantly altered (one-way ANOVA, *p* < 0.05) the time of immobility assessed in the tail suspension test (TST) (Figure 3). In this regard, plant chemical composition is influenced by the seasons, which lead to different biological properties [17]. Thus, the absence of a statistically significant decrease in the immobility time could be attributed to the low CGA and RU concentration in the extract of this collection [17]. However, it is not possible to rule out that other factors, such as the poor water solubility and physicochemical properties of non-encapsulated HEV, hindered its pharmacological action.

In this sense, producing a nano-based formulation containing HEV can be considered a promising alternative. As demonstrated in Figure 3, the animals that received a single NPHEV-10 administration at doses 5, 10, and 25 mg/kg presented a lower immobility time in comparison to the group that received the vehicle. Additionally, at these doses, the HEV nanoencapsulation provides an enhancement in antidepressant-like effect in comparison to the non-encapsulated extract. Remarkably, NPHEV-10 presented an antidepressant-like effect similar to fluoxetine (*p* < 0.05), a common antidepressant clinically used. As expected, the nanoparticles without the extract (NPB) had no effect in TST (*p* > 0.05). 

Reinforcing these data, the forced swimming test (FST) findings corroborated the antidepressant-like action demonstrated in the TST. Both encapsulated and non-encapsulated HEV reduced the immobility time in comparison to the control and NPB groups (Figure 4). However, while HEV had pharmacological effect only at the highest tested dose (50 mg/kg), the NPHEV-10 reduced the immobility time at all doses investigated (range 1 to 25 mg/kg). In this sense, the encapsulation enhanced HEV antidepressant-like action, which could be due to an improvement of limited physicochemical and stability properties of the extract. Notably, NPHEV-10 doses ranging from 5 to 25 mg/kg presented a similar effect to imipramine (20 mg/kg) (*p* < 0.05), indicating that the antidepressant-like effect of nanoencapsulated extract is comparable to classical antidepressant drugs.

Indeed, according to the literature, both CGA and RU present poor water solubility and oral bioavailability [28,47]. Thus, nanocarriers can improve compound bioavailability, either by increasing the permeability across the blood–brain barrier or by enhancing the dissolution. Likewise, Paczkowska and collaborators showed that the nanostructures (β-cyclodextrin inclusion complex) containing RU had improved solubility, stability, and permeability in comparison to non-encapsulated RU. Overall, an enhancement of antioxidant and microbiological actions was provided by associating RU with nanoparticles [47]. In addition, using Tween^®^ 80 in the nanoparticles composition is a promising strategy to enhance drug delivery through the blood-brain membrane [48]. 

Novel active substances that modulate various targets associated with depression disorder can be screened by preclinical behavioral tools, such as FST and TST [49]. Both behavioral tests present good predictive validity and provide rapid and economical detection of compounds with potential antidepressant-like effects [50]. However, the tests present different sensitivities to substances that modulate the neurotransmitter system, which could justify the distinct action profile of non-encapsulated HEV [51]. Thus, using both tests is an important strategy to better track the possible antidepressant-like effect of testing substances. Lastly, the treatment with NPHEV-10 caused no modifications (*p* > 0.05) in the animal spontaneous locomotion (Figure 5), discarding any possible psychostimulant effects that might be an antidepressant-like action indicative.

## 3. Materials and Methods

### 3.1. Chemicals

Methanol, ethanol, ascorbic acid, chlorogenic acid, sorbitan monooleate (Span 80^®^), and polysorbate 80 (Tween 80^®^) were obtained from Merck (Darmstadt, Germany). Eudragit^®^ RS100, DPPH radical, and RU were acquired from Sigma Aldrich Chemical Co. (St. Louis, MO, USA). Fluorescein (CAS No. 2320-96-9), trolox (CAS No. 53188-07-1), and AAPH (CAS No. 2997-92-4) were obtained from Aldrich (Milwaukee, WI). A stock solution of fluorescein (407 μmol/L) was prepared using potassium phosphate buffer (75 mmol/L; pH 7.4) and kept at 4 °C for ORAC assay. Both fluoxetine and imipramine hydrochloride were obtained from Galena^®^ (Porto Alegre, Brazil). 

### 3.2. Planta Material and Extract Preparation

*V. ashei* leaves (climax cultivar) were collected in Erechim, Rio Grande do Sul, southern Brazil (27°38′3″ S and 52°16′26″ W) in March 2014 and identified by V.S.C. A *voucher* specimen was registered by the number ICN 186814 at the Federal University of Rio Grande do Sul herbarium. The plant material was dried at 40 °C and grounded to obtain the plant material powder (80 µm). The HEV was prepared by maceration (60 g of powder) using a hydroalcoholic mixture extractor (water:ethanol; 1:1, *v*/*v*; 3 aliquots of 400 mL) for 72 h at room temperature [17]. The extract was submitted to reduced pressure to eliminate the solvent and dried by freeze-drying. The extract yield was 59.2% (*w*/*w*) and it was stored at 10 °C until further analysis. 

### 3.3. Analytical Method

The analytical method applied for sample evaluation was based on high-performance liquid chromatography (HPLC-UV/DAD). The Prominence Auto-Sampler (SIL-20A) equipped with Shimadzu LC-20AT (Shimadzu, Kyoto, Japan) pumps connected to a DGU-20A5 degasser and a CBM-20A integrator was used. An ultraviolet/ visible (UV-VIS) detector DAD SPD-M20A and software LC Solution 1.22 SP1 were applied. Phenomenex C_18_ column (4.6 mm × 250 mm; 5 µm) was used as the stationary phase. Injection volume was 40 µL and the gradient elution was conducted according to the scientific literature [17]. A previous study demonstrated that CGA and RU are the main phytochemical compounds extracted from the blueberry’s leaves; then the content of both compounds was estimated using the 365 and 327 nm wavelengths for RU and CGA detection, respectively [17]. 

### 3.4. Nanoparticle Suspensions Containing HEV 

Nanoparticles containing HEV were obtained by the interfacial deposition of the preformed polymer method [52]. Briefly, an amount of HEV (50, 100, or 250 mg), Eudragit^®^ RS100 (0.1%), and Span 80^®^ (0.077%) was dissolved in ethanol (50 mL). This phase was heated to 40 °C and kept under magnetic stirring for 60 min. An aqueous phase (50 mL) containing Tween 80^®^ (0.077%) was also prepared. After the solubilization of the constituents, the organic phase was injected into the aqueous phase and the mixture was stirred for 10 min. Following, the system was submitted to evaporation under reduced pressure to achieve 10 mL of the final volume, which corresponds to HEV concentrations of 5 mg/mL (NPHEV-5), 10 mg/mL (NPHEV-10), or 25 mg/mL (NPHEV-25). A formulation without HEV (NPB) was also prepared following the same procedures.

### 3.5. Physicochemical Characterization 

#### 3.5.1. Particle Size Analysis, Polydispersity Index, pH, and Zeta Potential

Photon correlation spectroscopy was used to determine the average particle size and the polydispersity index at 25 °C after diluting the samples in ultrapure water (1:500) (Zeta-sizer Nanoseries, Malvern Instruments, Malvern, UK). Zeta potential analyses were performed using the same instrument after diluting samples in 10 mM NaCl (1:500). The pH values were measured by immersing the electrode of a potentiometer in the nanoparticle suspensions. 

#### 3.5.2. Phenolic Content and Encapsulation Efficiency

The total RU and CGA content in nanocarriers were determined by diluting a nanoparticle sample aliquot in methanol (10 mL) and sonicating (10 min). Samples were then filtered through a 0.45 µm membrane and injected into the HPLC system (Section 3.3) for phytochemicals quantification [17]. The ultracentrifugation technique was performed to assess EE [53]. An aliquot of formulations (600 µL) was placed in a 10,000 MW centrifugal device (Amicon^®^ Ultra, Millipore, Burlington, MA, USA). Non-encapsulated phenolic compounds were separated from the nanoparticles by ultrafiltration (2200× *g*/10 min). The EE was calculated by the difference between the total and non-encapsulated compound concentrations, which was determined in the nanoparticles and in the ultrafiltrate, respectively, according to the equation:EE% = total content − free content/Total content × 100,(1)

### 3.6. Antioxidant Assays

#### 3.6.1. DPPH Assay

The DPPH test was applied to assess the radical scavenging effect [17]. The HEV, NPHEV-5, and NPHEV-10 were diluted with ethanol to achieve final concentrations of 25, 20, 15, 10, and 5 µg/mL. An aliquot of a DPPH ethanolic solution (0.3 mM) was added to the diluted samples and allowed to react at room temperature for 30 min prior to absorbance measurement (518 nm). Blank controls were prepared by mixing ethanol (1.0 mL) and diluted samples (2.5 mL), while negative control was composed of DPPH solution (1.0 mL) and ethanol (2.5 mL). The positive control was ascorbic acid. The DPPH radical inhibition percentage was calculated according to the equation:(2)$$\mathrm{AA}\%=100-\frac{\left(\mathrm{ABS}s-\mathrm{ABS}b\right)\times 100}{\mathrm{ABS}c}$$
where AA%: Antioxidant activity in percentage; ABS*s*: sample absorbance; ABS*b*: blank absorbance; ABS*c*: control absorbance. The results were demonstrated as the concentration (µg/mL) of the HEV or NPHEV required to neutralize 50% of DPPH free radicals (IC_50_). 

#### 3.6.2. Oxygen Radical Absorbance Capacity Test

The peroxyl radical neutralizing potential of the samples was determined by ORAC assay [17]. For this, HEV solution (25 mg/mL), NPHEV-5, NPHEV-10, and Trolox were diluted in phosphate buffer 75 mmol/L pH 7.4 to achieve a concentration range of 0–96 μmol/L. Then, an aliquot of fluorescein (150 μL; 81 nmol/L) was added to diluted samples (25 μL) in a 96-well plate and incubated at 37 °C for 10 min. Following, an aliquot of AAPH (152 mmol/L; 25 μL) was added. The fluorescence was determined (λ_exc_ = 485 nm and λ_em_ = 528 nm) every minute over 90 min using a SpectraMax M5 plate reader (Molecular Devices, San Jose, CA, USA). Results of the ORAC assay were calculated by linear regression relating Trolox concentrations with the net area under the kinetic fluorescein decay curve. The results were expressed as Trolox equivalents mmol/L. 

### 3.7. Acute Antidepressant-like Action Investigation

#### 3.7.1. Animals

Animals (Male adult Balb-C mice; 30 days old; 25–35 g) were obtained from Universidade Regional do Médio e Alto Uruguai e das Missões, Campus Frederico Westphalen (Brazil). Mice were housed in plastic cages (17 cm × 28 cm × 13 cm; five mice per cage) and kept under a 12 h light/dark cycle (lights on at 7 a.m.), constant temperature (22 ± 2 °C), and controlled humidity (60% relative humidity). Animals had free access to a standard diet and water. Mice received a single intragastric administration (ig, 10 mL/kg) of HEV (10, 25, and 50 mg/kg), NPHEV-10 (1, 2.5, 5, 10, and 25 mg/kg), fluoxetine (30 mg/kg), imipramine (20 mg/kg), or vehicles (e.g., 2% polysorbate 80 solution in saline or NPB), depending on the group that they belong to (8–10 animals/group). For treating the animals, HEV samples were prepared by weighting the extract and dissolving in 2% polysorbate 80 solution in saline. The NPHEV-10 was diluted using distilled water to achieve the desired concentrations prior injection in the animals. The doses, administration route, and general procedures for the treatment schedule were selected based on previous studies [54] and pilot experiments. 

#### 3.7.2. Tail Suspension Test 

Animals were suspended by the tail 60 cm above the floor using adhesive tape (1 cm from the tip of the end) in a light room. The time of immobility was recorded (in seconds) for 6 min by a blind observer concerning the treatments. Mice were considered immobile when they hung passively and completely motionless [55].

#### 3.7.3. Forced Swimming Test

The animals were individually placed in an acrylic cylinder (25 cm tall; 10 cm diameter) filled with distilled water (23 ± 1 °C). After the treatment (1 h), the animals performed the FST and the duration of immobility time (in seconds) was registered over 6 min by a blind observer regarding the experimental groups. The time of immobility was recorded when the mouse remained floating motionless or making only the movements necessary to keep its head above water [56]. 

#### 3.7.4. Locomotor Activity

The spontaneous locomotor skills were measured in the open-field test. Thus, mice were individually placed in a transparent acrylic box (40 × 30 × 30 cm) with the floor divided into 24 equal squares. Animals were habituated to the arena for 5 min and the number of crossings was recorded during the next 6 min [57].

### 3.8. Statistical Analysis

The results are presented as the mean ± standard deviation or mean ± standard error (SEM). Probability values less than 0.05 (*p <* 0.05) were defined as statistically significant. GraphPad Prism (version 6.0, GraphPad Software, San Diego, CA, USA) was employed for the one-way Analysis of Variance (ANOVA) followed by Newman-Keuls’ post hoc test for multiple group comparisons. 

## 4. Conclusions

In conclusion, our data demonstrated that the association of innovative technologies and natural product are a promising alternative to conventional therapies. For the first time, a *V. ashei* byproduct was nanoencapsulated in Eudragit^®^ RS100 nanoparticles. The formulation improved the antioxidant and antidepressant-like actions. Remarkably, the ORAC assay showed that nanoencapsulation increased almost 2-fold the antioxidant potential in comparison to the non-encapsulated extract. Furthermore, acute administration of NPHEV elicited superior antidepressant-like action in comparison to HEV and a similar effect of common antidepressant drugs used as positive controls (imipramine and fluoxetine), suggesting a new approach for treating depression. Future studies should be performed to investigate the antidepressant-like action in a chronic model, the mechanism involved in such effect, and the formulation’s safety and stability. As a complementary characterization, formulations can be evaluated using microscopic, thermal, and spectroscopic techniques. 

## Figures and Tables

**Figure 1 pharmaceuticals-16-00084-f001:**
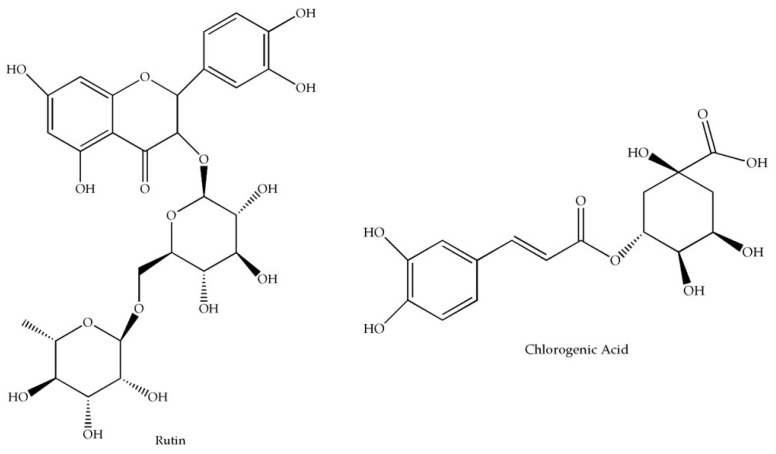
Chlorogenic acid and rutin chemical structures. The software ChemDraw 8.0 was used to draw the structures based on the PubChem chemistry database.

**Figure 2 pharmaceuticals-16-00084-f002:**
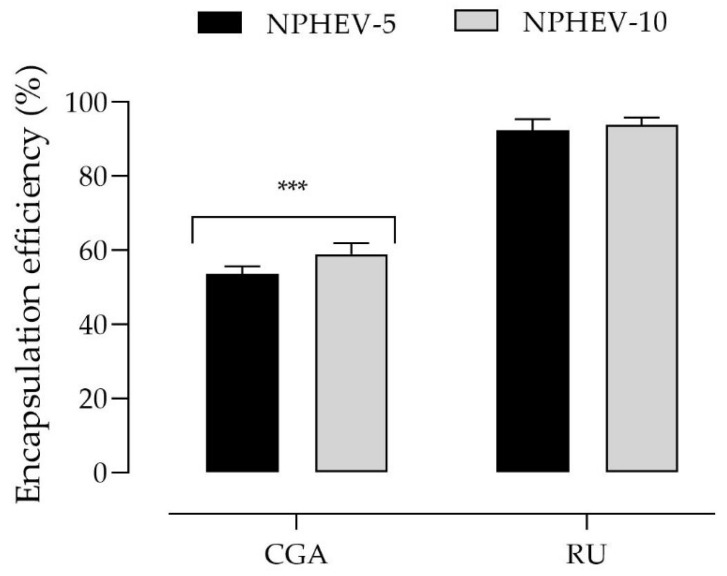
Chlorogenic acid (CGA) and rutin (RU) encapsulation efficiency (%). Asterisks denote a statistic difference between CGA and RU EE% (*** *p* < 0.001). Data were analyzed by ordinary one-way ANOVA, followed by Newman-Keuls’ tests.

**Figure 3 pharmaceuticals-16-00084-f003:**
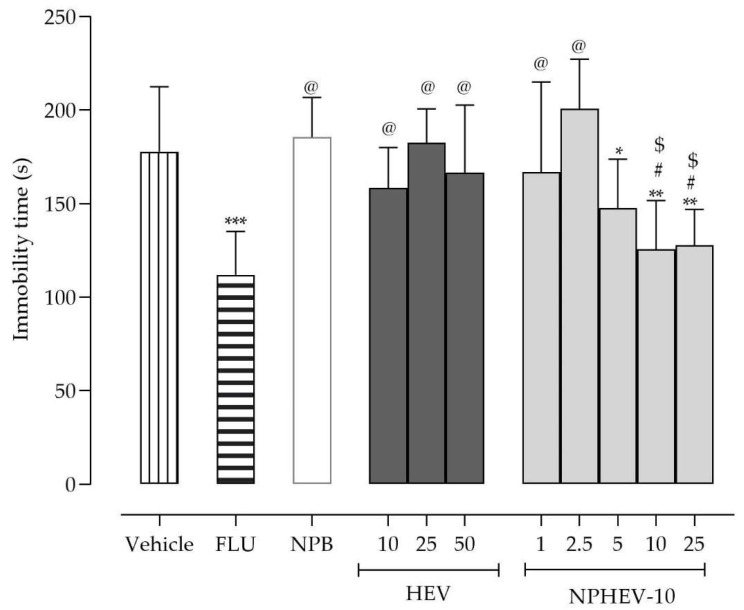
Dose–response curves of HEV (10, 25, and 50 mg/kg), NPHEV-10 (1, 2.5, 5, 10, and 25 mg/kg), NPB, and fluoxetine (FLU, 30 mg/kg) after single intragastric treatment in TST. Values are expressed as mean ± SEM. Asterisks denote the significant difference in comparison to the vehicle (* *p* < 0.05; ** *p* < 0.01; *** *p* < 0.001), sharps indicate the difference in comparison to NPB (^#^
*p* < 0.05), arroba denotes a significant statistical difference with fluoxetine group (^@^
*p* < 0.05), and ciphers denote the significant difference between the same doses of HEV and NPHEV-10 (^$^
*p* < 0.05). Data were tested using ordinary one-way ANOVA, followed by Newman-Keuls’ test.

**Figure 4 pharmaceuticals-16-00084-f004:**
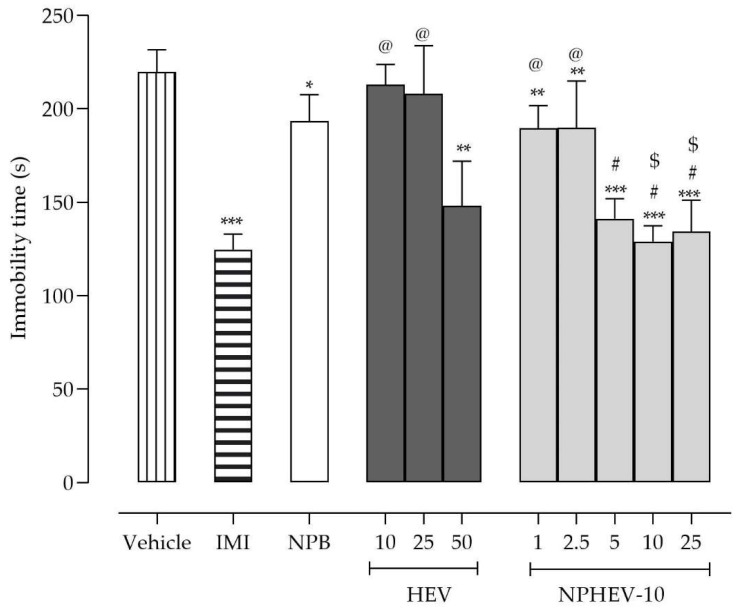
Dose–response curves of HEV (10, 25, and 50 mg/kg), NPHEV-10 (1, 2.5, 5, 10, and 25 mg/kg), NPB, and Imipramine (IMI, 20 mg/kg) after intragastric acute treatment in FTS. Values are expressed as mean ± SEM. Asterisks denote the significant difference in comparison to the vehicle (* *p* < 0.05; ** *p* < 0.01; *** *p* < 0.001), sharps indicate the difference in comparison to NPB (^#^
*p* < 0.05), arroba denotes a significant statistical difference with imipramine group (^@^
*p* < 0.05), and cipher denotes the significant difference between the same doses of HEV and NPHEV-10 (^$^
*p* < 0.05). All data were tested using ordinary one-way ANOVA, followed by Newman-Keuls’ test.

**Figure 5 pharmaceuticals-16-00084-f005:**
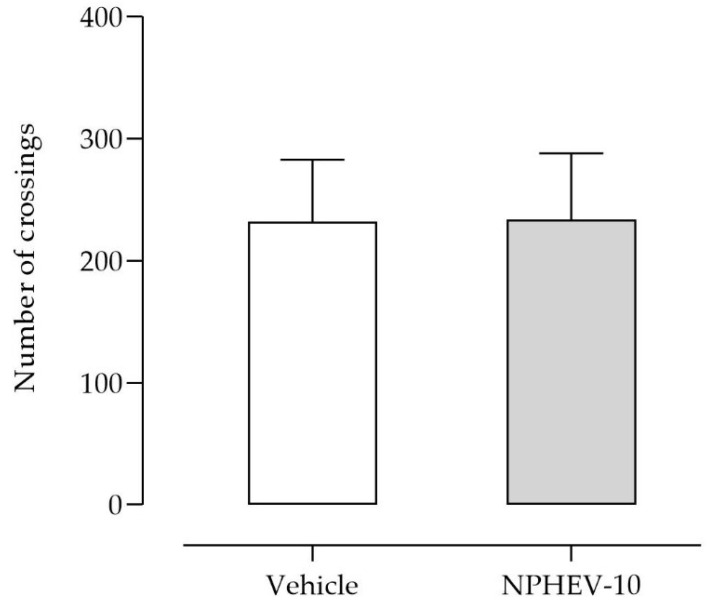
Spontaneous locomotor profile in the open-field test after the NPHEV-10 (25 mg/kg) administration. Data were tested using ordinary one-way ANOVA (*p* > 0.05).

**Table 1 pharmaceuticals-16-00084-t001:** Nanoparticle suspension physicochemical characteristics.

	Average Diameter (nm)	PDI	Zeta Potential (mV)	pH	CGA Content (%)	RU Content (%)
NPB	206 ± 7	0.163 ± 0.00	+8.8 ± 0.12	4.6 ± 0.03	-	-
NPHEV-5	143 ± 5 *	0.164 ± 0.03	+13.1 ± 10.3	3.9 ± 0.12	87.1 ± 5.9	92.3 ± 12.5
NPHEV-10	144 ± 1 *	0.142 ± 0.01	+15.5 ± 8.24	3.7 ± 0.10	91.7 ± 12.56	99.9 ± 8.4

Notes: PDI, polydispersity index; CGA, chlorogenic acid; RU, rutin; NPB, nanoparticles without HEV; NPHEV-5, nanoparticle suspension containing *V. ashei* leaf hydroalcoholic extract at 5 mg/mL; NPHEV-10, nanoparticle suspension of *V. ashei* leaf hydroalcoholic extract at 10 mg/mL. Asterisks denote a statistic difference with NPB, tested by ordinary one-way ANOVA, followed by Newman-Keuls’ tests.

**Table 2 pharmaceuticals-16-00084-t002:** HEV and NPHEV antioxidant action in DPPH and ORAC methods.

	DPPH (µg/mL) ^$^	ORAC Values (mmol/L)
HEV	12.39 ± 0.02	70 ± 4
NPHEV-5	16.30 ± 0.50 *	91 ± 8 *
NPHEV-10	14.40 ± 0.40 *^#^	133 ± 11 *^#^

Notes: HEV, *V. ashei* leaf hydroalcoholic extract; NPHEV-5, nanoparticle suspension containing *V. ashei* leaf hydroalcoholic extract at 5 mg/mL; NPHEV-10, nanoparticles suspension containing *V. ashei* leaf hydroalcoholic extract at 10 mg/mL. Values are expressed as mean ± standard deviation. Asterisks denote a statistical difference with HEV (* *p* < 0.05); hashtags denote the difference between NPHEV-5 and NPHEV-10 (^#^
*p* < 0.05). ^$^ DPPH results were expressed as IC_50._ Data were analyzed by ordinary one-way ANOVA, followed by Newman-Keuls’ tests.

## Data Availability

Data is contained within the article.

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
