# Peer review of "Nanoencapsulation of Vaccinium ashei Leaf Extract in Eudragit® RS100-Based Nanoparticles Increases Its In Vitro Antioxidant and In Vivo Antidepressant-like Actions"

_pharmaceuticals, 2023, doi:10.3390/ph16010084_

Round 1

Reviewer 1 Report

The manuscript “Nanoencapsulation of Vaccinium ashei leaf extract in Eudragit® RS100-based nanoparticles increases its in vitro antioxidant and in vivo antidepressant-like actions” requires improvement. Authors should revise and improve the English language and grammar throughout the manuscript.

Introduction:

Line 41-42: Please add a recent statistic

Line 68: Once Vaccinium ashei has been introduced, abbreviation V. ashei (italic) should be used in the following text. Please revise throughout the manuscript.

Results and Discussion:

Please avoid using “we”, “us” in the manuscript.

Line 93-96: Integrate these statements into Section 2.1

Line 219: Please be accurate to use NPHEV-10. Revise accordingly throughout the text.

Materials and Methods:

Please include the name of the botanist who identified the plant in Section 3.2

Please include a reference in Section 3.5.2 and 3.6.1

Line 337-339: Please check again the abbreviations used in the formula and the note.

Conclusions:

Authors are advised to rewrite the conclusion to provide a comprehensive summary and significant findings of the study. Also, future works in specific should be included as well.

References:

Please adhere strictly to the Journal’s requirement. Inconsistencies and some references did not put Journal’s name. Eg. #12.

Please include a proof of ethical approval from the related authority to conduct the animal works in the ethical statement.

The use of abbreviation is not consistent. Once abbreviation is introduced in full, it should be used in abbreviated forms throughout the manuscript. Please carefully check and revise thoroughly.

Include the structures of chlorogenic acid and rutin in the manuscript.

Reviewer 2 Report

In the manuscript titled “Nanoencapsulation of Vaccinium ashei leaf extract in Eudragit® RS100-based nanoparticles increases its in vitro antioxidant and in vivo antidepressant-like actions”, the antioxidant and antidepressant-like actions were investigated. This research work is well-established and supported by the results. After addressing the following questions appropriately, the manuscript could be accepted for publication.

1.   There are a lot of typo errors in the manuscript. For example, line 38 mental disorders diseases, line 102 lipossomes, line 120 diference, line 145 polimeric, line 176 electrondonating, line 196 compouns, line 212 vheicle, line nos. 212, 214, 224, 240, 241, 242, 247, 251, 276, 299, etc

2.   Scientific name of the plant should be uniformly represented all over the manuscript. Sometimes, it was mentioned as Vaccinium ashei and sometimes as Vaccinium ashei. For example, lines 118, 119, 167, 168, etc.

3.   Consider rewriting the following statement:

Line 126, “Additionally, drug loading and release and nanoparticle physicochemical stability can also be impacted bt those properties”.

Line 132: “Furthermore, the suspensions presented acidic pH values (3.9 ± 0.12 and 3.7 ± 0.10), similar to other nanostructured systems made of Eudragit® RS100”.

Line 180: “Both are spectrophotometric tests, which can cause a selectivity lacks due to spectral interference; in this case, the using tests with different mechanisms provide greater reliability in the results”.

4.   What is the volume of the sample used to determine the particle size and PDI? For example, if the sample volume is 1 ml, you take about 2 µl of the nanoformulation and dilute it with about 980 µl of ultrapure water. In this study, the sample was highly diluted (1:500), are you sure there are enough nanoparticles in the sample solution to adequately determine?

5.   Line 319, is the filter 45 µm or 0.45 µm?

6.   The conclusions section can be improved supported by the study's results.

Reviewer 3 Report

The research work" Nanoencapsulation of Vaccinium ashei leaf extract in Eudragit® RS100-based nanoparticles increases it's in vitro antioxidant and in vivo antidepressant-like actions" is covered under this scope of the journal. 

The observation comments are:

1. Use of nanoparticles at many places in place of nanocapsules. Authors should be firm about terms and meanings. 

2. Why Eudragit® RS100 selected as polymer? Give the rationale with citations in the text. 

3. Material method should be 2.0 and the result discusses 3.0. Please rearrange. 

4. Avoid using we, us, I, etc. 

5. Line: The encapsulation efficiency (EE) demonstrated that almost 100% of RU was entrapped within the polymeric matrix of the particles (92 and 94% for NPHEV-5 and NPHEV-10, respectively). Is authors confirm this, as it is difficult to get such high EE, especially when the drug is water soluble. 

6. Is a stability study of formulation performed? if yes show the results. 

7. SEM of nanocapsules? 

Round 2

Reviewer 2 Report

Acceptable for publication

Reviewer 3 Report

Thank you for your rebuttal.